# Development and Internal Validation of a Model for Predicting Overall Survival in Subjects with MAFLD: A Cohort Study

**DOI:** 10.3390/jcm13041181

**Published:** 2024-02-19

**Authors:** Caterina Bonfiglio, Angelo Campanella, Rossella Donghia, Antonella Bianco, Isabella Franco, Ritanna Curci, Claudia Beatrice Bagnato, Rossella Tatoli, Gianluigi Giannelli, Francesco Cuccaro

**Affiliations:** 1National Institute of Gastroenterology—IRCCS ‘S de Bellis’, 70013 Castellana Grotte, BA, Italy; angelo.campanella@irccsdebellis.it (A.C.); rossella.donghia@irccsdebellis.it (R.D.); antonella.bianco@irccsdebellis.it (A.B.); isabella.franco@irccsdebellis.it (I.F.); ritanna.curci@irccsdebellis.it (R.C.); claudia.bagnato@irccsdebellis.it (C.B.B.); rossella.tatoli@irccsdebellis.it (R.T.); gianluigi.giannelli@irccsdebellis.it (G.G.); 2Local Health Unit—Barletta, Andria-Trani, 76121 Barletta, Italy; francescocuccaroepi@gmail.com

**Keywords:** prognostic index, MAFLD, mortality

## Abstract

**Background & Aims**: Fatty liver disease with metabolic dysfunction (MAFLD) is a new concept proposed to replace the previous concept of Non-Alcoholic Hepatic Steatosis (NAFLD). We developed and internally validated a prognostic model to predict the likelihood of death in a cohort of subjects with MAFLD. **Methods:** Our work involved two steps: the first was the construction of a bootstrapped multivariable Cox model for mortality risk prognosis and the second was its validation. **Results:** The study cohort included 1506 subjects, of which 907 were used for internal validation. Discriminant measures for the final model were R^2^_D_ 0.6845 and Harrell’s C 0.8422 in the development and R^2^_D_ 0.6930 and Harrell’s C 0.8465 in the validation. We used the nine independent prognostic factors selected by the LASSO Cox procedure and fitted by the bootstrap Cox survival model, and observed β were: Gender 0.356 1.42 (*p* < 0.008), Age 0.146 (*p* < 0.001), Glycemia 0.004 (*p* < 0.002), Total Cholesterol −0.0040 (*p* < 0.009), Gamma Glutamyl Transpeptidase 0.009 (*p* < 0.001), SBP 0.009 (*p* < 0.036), DBP −0.016 (*p* < 0.041), ALP 0.008 (*p* < 0.071) and Widowhood 0.550 (*p* < 0.001). **Conclusions:** We produced and validated a model to estimate the probability of death in subjects with MAFLD. The instruments we used showed satisfactory predictive capabilities.

## 1. Introduction

Metabolic dysfunction associated with fatty liver disease (MAFLD) is a new concept proposed in 2020 to move the diagnostic entity from a “non-condition” to an inclusive disease [1]. A MAFLD diagnosis does not necessarily include other etiologies of liver disease, such as excessive alcohol consumption or viral hepatitis [2]. The MAFLD diagnosis is built on a hallmark sign such as hepatic steatosis, plus one of the following three metabolic conditions: overweight/obesity (Subtype 1), evidence of metabolic dysregulation (MD) in lean subjects (Subtype 2) or diabetes mellitus (Subtype 3) [3].

Interestingly, it has yet to be ascertained if the new definition provides a better prognosis of endpoints such as mortality. Although a large body of scientific evidence supports the association of MAFLD with cardiovascular diseases (CVD), malignancies, and liver-related endpoints, its impact on mortality is still debatable [4].

The inclusion criteria of the new MAFLD terminology identify a group of people with fatty liver but also with metabolic dysregulation that would not have been identified if we had only selected those with NAFLD. Subjects with MAFLD are more likely to have advanced fibrosis and higher rates of overall mortality than those with NAFLD [4].

It should be noted that patients with MAFLD have some of the following risk factors for death, whose importance should not be underestimated: hepatic steatosis, obesity, diabetes, metabolic syndrome and alcohol abuse [5,6].

New studies have found that metabolic dysfunction associated with fatty liver disease (MAFLD) is associated with the prevalence of chronic kidney disease (CKD). However, it is still unknown if MAFLD is associated with the development of CKD and the incidence of end-stage renal disease (ESKD). One study sought to clarify the association between MAFLD and incident ESKD in the UK Biobank prospective cohort [7].

Other studies have found that patients with MAFLD are at greater risk of early or subtle cognitive dysfunction than healthy individuals, but this relationship was not correlated with the presence of metabolic syndrome, as cognition is primarily regulated by domains of visuospatial and executive function associated with the prefrontal cortex [8].

Prognosis research in healthcare forecasts future outcomes in those with a disease or health condition. The aims and findings from prognosis research studies can be summarized as the average risk or value of an outcome among those with the health condition of interest in a particular setting [9].

Usually, in prognostic research, multiple variables are used to make the predictions as accurate as possible, and to ultimately institute preventive measures that will act against the occurrence of the outcome. This implies that although a prognostic model can provide evidence of the causality or pathophysiology of the outcome being studied, this is neither a goal nor a requirement. Then, all variables potentially associated with the outcome can be considered in a prognostic study [10].

Therefore, the aim of the present cross-sectional study was to develop and internally validate a multivariable model to predict the probability of death in a cohort of subjects diagnosed with MAFLD.

The study was reported according to the guidelines “Transparent reporting of a multivariable prediction model for individual prognosis or diagnosis” (TRIPOD) (Appendix B).

This model should undergo external validation, consisting of both discrimination and calibration, before being employed in clinical or research practice.

## 2. Materials and Methods

### 2.1. Study Population

Details about the study population have been published elsewhere [11,12,13].

Briefly, two different prospective cohort samples from two studies conducted by the Laboratory of Epidemiology and Biostatistics of the National Institute of Gastroenterology at the Research Hospital IRCCS ‘Saverio de Bellis’ (Castellana Grotte, Bari, Italy) were included. The MICOL Study is a cohort study started in 1985 and was followed up in 1992, 2005–2006 and 2013–2016. In 2005–2006 this cohort was supplemented with a random sample of subjects (PANEL study) who were 30–50 years-of-age, to compensate for the ageing of the cohort.

The NUTRIHEP study is a cohort extracted from the medical records of general practitioners in Putignano (≥18 years), started in 2005. Using a systematic random sampling procedure, a sample of the general population aged ≥ 18 years was extracted from the list of GP registers. We used the GP registers rather than an extraction from the census, because no significant differences were found between the age and gender distribution of the general population of Putignano and those registered in the GP registers. The law in Italy requires everyone to have a general practitioner, and so the list of the general population in the GP offices and the census correspond.

During the follow-up visits, the participants underwent all the assessments required by the protocol. On an annual basis the mortality of the two cohorts, including the causes of death, was updated with data extracted from the regional register and electronically linked to our database.

The recruited sample consisted of 1675 subjects aged >30 years (543 females and 1132 males) diagnosed with MAFLD among participants in the second follow-up of the MICOL cohort (from May 2005 to January 2007); and those in the NUTRIHEP cohort, recruited from July 2005 to January 2007, and observed until 31 December 2022.

In the development cohort, we limited the analysis to 1506 subjects who were, at maximum, 86-years-old when they died (life expectancy age for the Apulia region) [14].

Using the criteria of Riley et al. [15,16] a sample size of 907 (60.23%) subjects was calculated, which was more than sufficient to estimate an R-square (Cox-Snell) of 0.14, with an overall mortality rate of 0.0111 estimated in the development cohort. The sub-cohort comprised 60% of the initial cohort and had uniformly distributed random variances [17] (Appendix A).

The studies were conducted at the National Institute of Gastroenterology, IRCCS “S. De Bellis”, in Castellana Grotte (Bari, Italy).

All procedures were performed according to the ethical standards of the institutional research committee (National Institute of Gastroenterology, IRCCS “S. De Bellis” Research Hospital), after the ethical committee approved the MICOL Study (DDG-CE-589/2004 18 November 2004) and the NUTRIHEP Study in 2005 (DDG-CE-502/2005 20 May 2005). The study was conducted in accordance with the 1964 Helsinki declaration and later amendments, and written informed consent was obtained from each participant.

### 2.2. Data Collection

At the baseline, trained personnel interviewed participants to collect information about sociodemographic characteristics (including educational level, work, and marital status), health status, personal history, and history of tobacco use. Standard procedures were used to measure weight and height. Weights were taken on an electronic balance, SECA^®^, and recorded to the nearest 0.1 kg. Height was measured with a wall-mounted SECA^®^ stadiometer and recorded to the nearest 1 cm. Blood pressure and Body Mass Index (BMI) were calculated by following international guidelines [18]. The average of 3 blood pressure measurements was calculated. A fast venous blood sample was drawn for each participant and processed according to standard laboratory techniques in our central laboratory.

All subjects underwent a standardized ultrasound examination, using a Hitachi H21 Vision device (Hitachi Medical Corporation, Tokyo, Japan) and a 3.5 MHz transducer.

A scoring system was adopted to obtain a semi-quantitative evaluation of fat in the liver [19]. Steatosis was dichotomously classified as absent (Score 0) vs. present (Score ≥ 1).

### 2.3. Tracing Procedures and Outcome Assessment

The vital status of participants at the end of the study was obtained from the municipalities of Castellana Grotte and Putignano and electronically linked to the database. Inquiries were also made at the municipalities of current residence about subjects who had moved. Causes of death were extracted from the Apulian Regional Registry, using the death certificate, as established by WHO guidelines [20].

### 2.4. Predictive Factors

Candidate variables for the final prognostic model included Gender (0 Female, 1 Male), Enrollment Age (age, years), Smoking Habit (0 Never or Former, 1 Current), Systolic Blood Pressure (SBP) (mmHg), Diastolic Blood Pressure (DBP) (mmHg), DBP (mmHg), Glucose (mg/dL), Total Cholesterol (TC, (mg/dL)), High-Density Lipoprotein Cholesterol (HDL-C, (mg/dL)), Low-Density Lipoprotein Cholesterol (LDL-C, (md/dL)), Triglycerides (TG, (mg/dL)), Alanine Amino Transferase (ALT, (U/L)), γ-Glutamyl Transferase (GGT, (U/L)), Aspartate Amino Transferase (AST, (U/L)), Alkaline Phosphatase (ALP (U/L)), Widowed Status (1 Widowhood, 0 Otherwise), Education (0 Primary School, 2 Secondary School, 3 High school, 4 Graduate, 5 Illiterate), Olive Oil Consumption (gr/die), Wine Consumption (ml/die), Beer Consumption (ml/die) and Spirit Consumption (ml/die).

### 2.5. Missing Data

No data were missing.

### 2.6. Statistical Analysis

Data are presented as mean (±SD), median (IQR) and frequency (%). Since they were not normally distributed, continuous variables are reported as medians (50th percentile) and interquartile range (25th and 75th percentiles).

Discrete variables are reported as frequency and percentage (%).

Overall Survival (OS) in MAFLD patients was the primary outcome of interest. We considered time-to-event from the baseline date to death, moving out of the area, or the end of the study (31 December 2022).

No subjects were lost during the follow-up and so there are no censored data.

Survival curves were estimated using the Kaplan–Meier method, and the least absolute shrinkage and selection (LASSO Cox) procedure was adopted to reduce the number of candidate predictors and select those most helpful in constructing the prognostic model [21]. Selection was performed on the basis of statistical and subject matter considerations.

A bootstrapped multivariable Cox survival model was fitted, and proportional hazards were tested using Schoenfeld residuals.

The model performance and the discriminatory ability in both the development and validation cohorts were probed using the C-index and R^2^_D_ index [22]. Calibration Models were then applied to verify the agreement between predicted and observed probabilities [23,24,25].

We used the tertiles obtained from the PI to create a Kaplan–Meier graph.

The Stata (version 18.0) statistical package was used to perform all statistical analyses (StataCorp, 4905 Lakeway Drive, College Station, TX, USA). The user-written programs *Lasso cox*, *stcox*, *pmsampsize*, and *stcoxcal* were used.

### 2.7. TRIPOD Guidelines

To construct the predictive model, guidelines for the transparent reporting of a multivariable prediction model for individual prognosis or diagnosis (TRIPOD) were followed [26]. Appendix B contains the item checklist as required by the TRIPOD guidelines.

## 3. Results

### 3.1. Cohorts Characteristics

In total, 1506 subjects with MAFLD aged less than 86-years-of-age at death (equal to the life expectancy of Puglia) were included in the development cohort, and 907 of the 1506 were in the internal validation cohort (Figure 1).

The baseline characteristics of the subjects in the development and validation cohorts are detailed in Table 1.

During a median follow-up of 16.87 years (IQR 16.12; 17.22), 271 deaths (18.0) occurred (Table 1). Of these, 102 were cancer-related (10 liver cancer); a further 65 were due to cardiovascular disease (CVD), 20 to digestive diseases (15 liver cirrhosis) and 84 to other causes, of which 20 deaths were attributable to respiratory disorders and 20 to mental and behavioural disorders.

Of the cohort of 1506 subjects, the mean age of the population still alive at the end of the study was 67.6 years (±10.30); the average age at death was 74.8 years (±9.63), 76.1 (±9.62) for women and 74.20 (±9.60) for men.

It was observed that only 22 subjects used drugs that can induce fatty liver disease (e.g., corticosteroids, valproic acid, amiodarone, methotrexate, tamoxifen, atypical neuroleptics, tetracycline).

### 3.2. Selection of Prognostic for Multivariable Modeling

The LASSO Cox regression model [27], with λ_CV_ at 0.032, identified nine prognostic factors: Gender (Male versus Female). Enrollment Age, Glucose, Total Cholesterol, γ-Glutamyl Transferase, Systolic Blood Pressure; Diastolic Blood Pressure, Alanine Amino Transferase and Widowhood (Appendix A).

### 3.3. Construction of the Multivariable Prognostic Model

These nine predictors were entered into the Cox survival model to develop a prognostic multivariable model. Appendix A shows the test of the proportional hazards assumption, while Appendix A shows the effect that removing variables from the Cox model has on the R^2^_D_ and C-index of the final model.

The most explanatory variable in our model was Enrollment Age followed, in descending order, by GGT, SBP, Glucose, Gender, DBP, TC, ALP and Widowhood. The discriminant measures of the model were: R^2^_D_ 0.6845 (SE 0.03) and C-index 0.8422.

Figure 2 shows the development cohort Kaplan–Meier curves and the predicted mean survival curves. It can be seen very clearly that the predicted curves are very close to the observed survival curves.

Cox multivariable model selection, with 1000 bootstrap samples based on the development cohort, is shown in Table 2.

### 3.4. Internal Validation

Internal validation of the model’s discriminatory ability yielded a C-index of 0.8465, demonstrating the model’s good discriminatory performance. The calibration slope was 1.18, reflecting a satisfactory calibration (Figure 3). Some miscalibration in the large is evident, with underprediction of event probabilities in the validation dataset.

Table 3 compares discriminant measures evaluated in the development and validation cohorts.

### 3.5. The Prognostic Model Risk

Based on the nine independent prognostic factors selected by the LASSO Cox model and fitted by the bootstrapped Cox survival model, the Prognostic Index (PI) was = [0.146 × (age years) + 0.356 × (Gender) + 0.004 × (Glucose mg/dL) +0.009 × (SBP mmHg) − 0.016 × (DBP mmHg) − 0.004 × (TC mg/dL) + 0.009 × (GGT U/L) + 0.008 (ALP U/L) + 0.550 × (widowhood)].

Variable coding: *Gender: (0 Female; 1 Male); Widowhood: (1 Widow/er; Otherwise 0)*

### 3.6. Clinical Examples

Two clinical application examples based on the prognostic prediction obtained:A woman, aged 30 years-of-age, with Glucose 100 (mg/dL), TC 245 (mg/dL), GGT (U/L) 38 ALP (U/L) 41, SBP (mmHg) 145, DBP (mmHg) 95 and not a widow, at least 16 years since the diagnosis of MAFLD.

PI = [0.146 × (30 years) + 0.356 × (0) + 0.004 × (100 mg/dL) + 0.009 × (145 mmHg) − 0.016 × (95 mmHg) − 0.004 × (245 mg/dL) + 0.009 × (38 U/L) + 0.008 (41 U/L) + 0.550 × (0) = 4.25

The second example:2.A man, aged 63 years-of-age, with Glucose 193 (mg/dL), TC 171 (mg/dL), GGT (U/L) 39 ALP (U/L) 78, SBP (mmHg) 140, DBP (mmHg) 70 and a widower, at least 16 years since the diagnosis of MAFLD.

PI = [0.146 × (63 years) + 0.356 × (1) + 0.004 × (193 mmol/L) + 0.009 × (140 mmHg) − 0.016 × (70 mmHg) − 0.004 × (171 mmol/L) + 0.009 × (39 μkat/L) + 0.008 × (78 μkat/L) + 0.550 × (1) = 11.31

Using the obtained equation, we calculated the prognostic index for the 1506 subjects in the development cohort and divided it into tertiles: PI < 8, 8 ≤ PI ≤ 10 and PI > 10.

In each tertile there are 502 subjects. In the first group 25 deaths occurred, in the second 79 and the third 164.

We represented the Kaplan–Meier curves graphically, using the categories of tertiles obtained.

The log-rank test showed the statistically significant diversity of the three categories, (*p* < 0.001) confirming a different survival for subjects with MAFLD after the 17-year follow-up (Figure 4).

## 4. Discussion

An ideal prognostic model should be easy to use, only include the most relevant characteristics of each subject that are related to the disease of interest, and accurately distinguish groups of subjects with different prognoses [28]. Our model meets the first two criteria and has a good discriminatory ability, although there is still room for improvement. The prognostic index is based on nine predictors that readily available in routine clinical work.

The prognostic factors in subjects diagnosed with MAFLD that were included in the final model are: Gender, Age, Glucose, TC, GGT, SBP, DBP, ALP and Widowhood. Internal validation showed a good discriminatory ability, with a C-index of 0.8465, and explained the relative log risk scale variance of the D statistic (R^2^_D_) of 0.6930 and satisfactory calibration (slope 1.18).

The highest risk of death was observed in older participants. Although our cohort had a median exposure of 16.87 years (IQR: 16.12; 17.22) to MAFLD-related risk factors (obesity, diabetes, hypertension, hypercholesterolaemia, etc.), this was not associated with a high mortality rate in any age group.

Our prognostic equation includes both non-modifiable variables, such as age and widowhood, and modifiable variables, such as TC, GGT, ALP, SBP, DBP and Glucose, which can be improved by a healthy diet (e.g., the Mediterranean diet [22,23]) and/or taking specific drugs [24,25]. The model was built using a cohort from two towns in Southern Italy, where a diet linked to the traditions of Mediterranean cuisine is common. Oil and wine are locally produced, and so are vegetables and dairy products. The olive oil is made using the olives from the countryside which, harvested between November and January [29]. Extra virgin olive oil is organic and subject to a controlled supply chain.

Our hypothesis is supported by the literature, and more specifically by the multiple papers that show the benefits of the Mediterranean diet in helping to prevent metabolic diseases, cancers, and cardiovascular disease [30,31], including a recent study that showed the protective effect of extra virgin olive oil on overall mortality and NAFLD [32].

Indeed, a distinctive gut microbial dysbiosis was recently observed to be associated with chronic alcoholic fatty liver disease and MAFLD in an animal model [33]. It has been suggested that the gut microbiota may modulate the protective association between the Mediterranean diet, the most common diet in this region, [34] and a lower risk of cardiometabolic disease [35,36].

Moreover, olive oil has always been essential to the Mediterranean diet, especially in the older population. As part of the Mediterranean diet, it is associated with human health benefits, especially for the Cardiovascular system, and countering obesity, diabetes and related metabolic disorders. Olive oil has a high phenol content, and has been shown to positively affect oxidised LDL, conjugated dienes, and hydroxy fatty acids, and also decrease the LDL versus HDL ratio [37,38,39].

Although the effect of alcohol remains controversial, epidemiological studies have shown that regular, moderate consumption of wine (one to two glasses per day) is associated with a lower incidence of cardiovascular disease (CVD), hypertension, diabetes and some types of cancer [40,41].

Recent studies have shown that population-level health risks are even associated with low levels of alcohol consumption, and are found to vary from region to region and be more pronounced in younger than older populations [42].

Moderate amounts of alcohol, mainly red wine, are part of the Mediterranean dietary pattern, and its effect on longevity has been extensively studied [43] in Mediterranean countries where it is mainly consumed during meals.

We observed a different trend between age groups in the daily consumption of grams of olive oil, seasonal fruit, locally produced vegetables, and the amount of wine drunk, demonstrating different eating habits. We observed that older classes have a higher daily consumption of vegetables and oil than younger classes (Appendix A).

This is a methodological work, applied to a cohort of subjects with MAFLD in two cities in Apulia, where the traditions of the Mediterranean Diet have been established for decades. All the results obtained in this study are taken from data meticulously collected by the Laboratory of Epidemiology of the National Institute of Gastroenterology (in the ‘S de Bellis’ Research Hospital, Castellana Grotte, (BA)) during the two cohort studies that have already been described in detail.

We regard this as a pilot study on the prediction of mortality in subjects with MAFLD, as there is nothing similar in the literature.

The prognostic index obtained from our study is easy for clinicians to use because readily available clinical and demographic variables were employed.

We consider this study to be very important because of the mortality risks associated with continuing MAFLD over time, as not all subjects may benefit from the positive effects of the Mediterranean diet [4,5,7,8].

Finally, we included two clinical examples to show the feasibility of the prognostic mortality risk equation and tertiles, which were obtained from the PI, being used to draw a Kaplan–Maier graph to hypothesize three risk groups.

### Strengths and Limitations

Our study has several strengths, particularly the cohort design and the sizable random population sample from a geographic area where the Mediterranean diet is widespread. Moreover, the same operators performed the complete exposure assessment (biochemical, ultrasonography) in our hospital, ensuring a high reproducibility of measurements. A potentially serious limitation is the absence of measures of physical activity, which is particularly important because previous research results link physical activity to all-cause and cause-specific mortality. The absence of measurements of glycosylated haemoglobin and information on individuals with HCV and HBV could also be a problem, as an over- or underestimated effect of diet could occur because of a confounding effect or modification of physical activity. However, selection bias is unlikely to be present, and classification bias, if present, should be non-differential and would only produce a bias toward the null hypothesis [44]. Limitations include a discrete calibration between observed and expected values, which may be due to the way the group was formed for validation and the discrepancies between the predicted probability and the actual underlying risk for each individual, which cannot be observed [45].

## 5. Conclusions

We report the building and internal validation of a simple prognostic model to estimate the probability of death in subjects with MAFLD. The model was built using a cohort from two towns in southern Italy, where a diet linked to the traditions of Mediterranean cuisine is common.

The prognostic factors in subjects diagnosed with MAFLD included in the final model are Gender, Age, Glucose, TC, GGT, SBP, DBP, ALP and Widowhood. Internal validation showed a good discriminatory ability, with a C-index of 0.8465 and explained variance on the relative log risk scale of the D statistic (R^2^_D_) of 0.6930 and satisfactory calibration (slope 1.18).

This is a pilot study addressed to the prediction of mortality in subjects with MAFLD, whose value and contribution is further underlined by the absence of similar works in the literature.

The instruments we chose showed good predictive capabilities.

We used the tertiles, obtained from the PI, to draw a Kaplan–Maier graph to hypothesise three risk groups.

This model should undergo external validation before being used in clinical or research practice. External validation is necessary to determine a prediction model’s reproducibility, and also establish its generalizability to new and different patients [46].

## Figures and Tables

**Figure 1 jcm-13-01181-f001:**
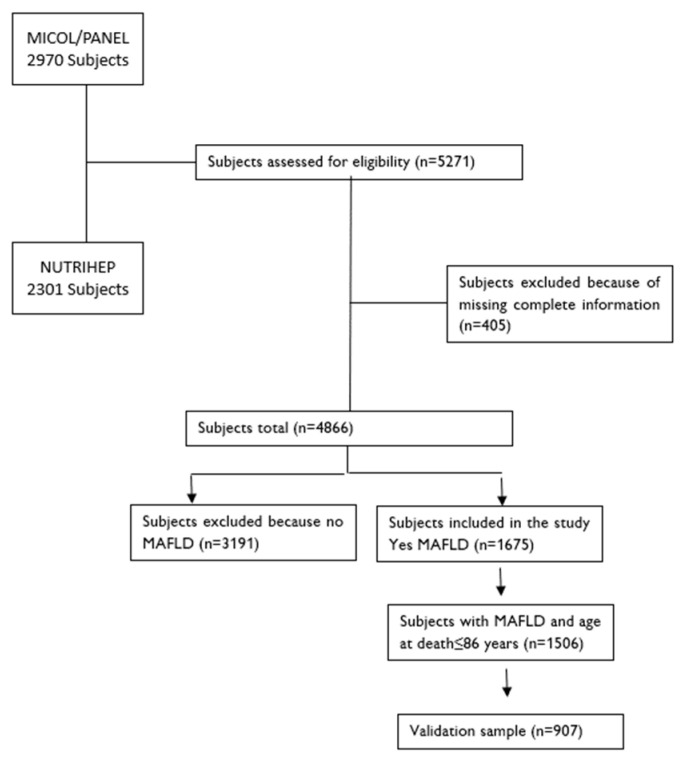
Participant flow.

**Figure 2 jcm-13-01181-f002:**
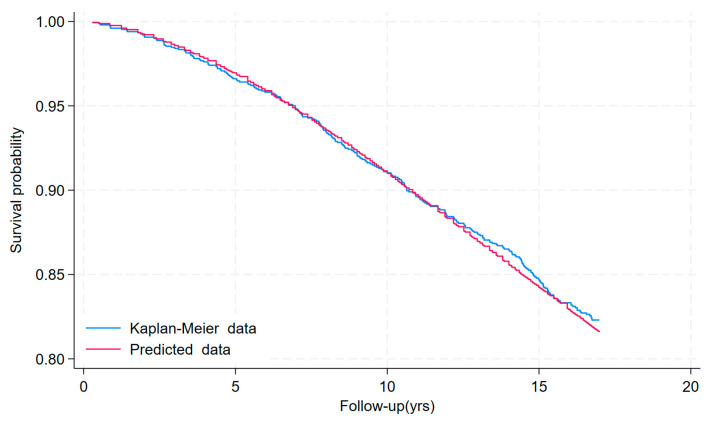
Kaplan–Meier curves and predicted mean survival curves in the development cohort.

**Figure 3 jcm-13-01181-f003:**
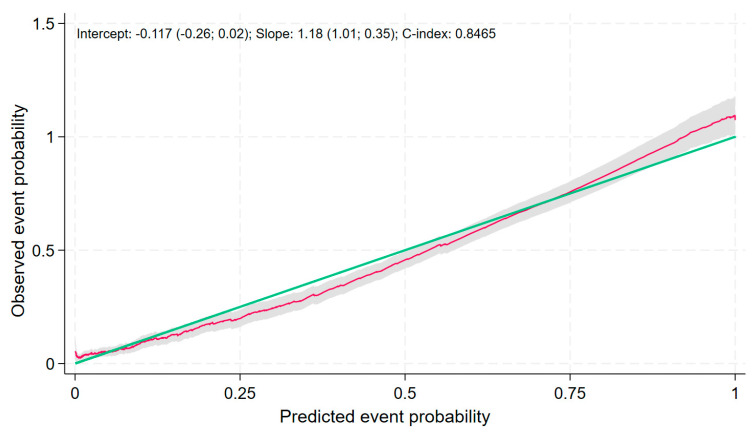
Calibration plot for the survival prediction model in the validation cohort. The red line shows the pseudo-values at the time of 17 years. The green line shows the perfect calibration. In gray the 95% CI.

**Figure 4 jcm-13-01181-f004:**
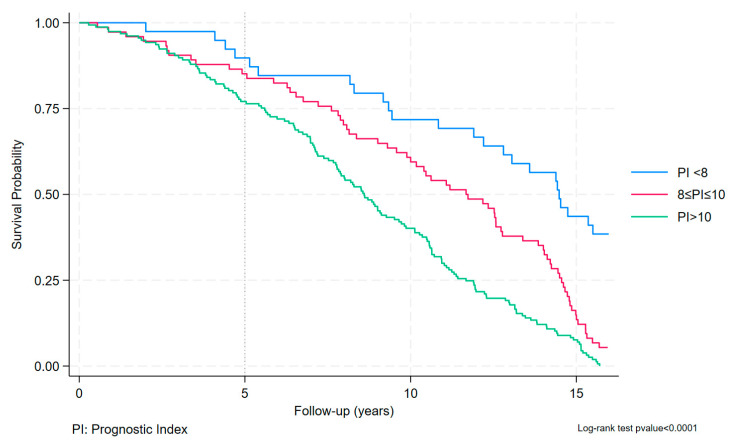
Kaplan–Meier curves for the development cohort by prognostic index groups.

**Table 1 jcm-13-01181-t001:** Baseline characteristics of the development and validation cohorts.

	Development	Validation
N	1506	907
Gender ^a^		
Female	469 (31.1)	298 (32.9)
Male	1037 (68.9)	609 (67.1)
Enrollment Age ^c^ (years)	53.25 (11.48)	53.25 (11.50)
SBP ^b^ (mmHg)	127.80 (17.19)	128.33 (17.47)
DBP ^b^ (mmHg)	80.00 (70.00; 85.00)	80.00 (70.00; 85.00)
Weight ^b^ (kg)	82.00 (74.00; 91.00)	82.00 (74.00; 91.00)
BMI ^b^ (kg/m^2^)	30.13 (27.59; 33.50)	30.25 (27.73; 33.53)
Glucose ^b^ (mg/dL)	107.00 (99.00; 117.00)	107.00 (100.00; 117.00)
TG ^b^ (mg/dL)	137.00 (92.00; 189.00)	136.00 (92.00; 189.00)
TC ^b^ (mg/dL)	202.00 (177.00; 232.00)	201.00 (176.00; 233.00)
HDL-C ^b^ (mg/dL)	45.00 (39.00; 52.80)	45.00 (39.00; 52.80)
LDL-C ^b^ (mg/dL)	126.00 (103.50; 149.10)	123.00 (103.00; 149.90)
ALT ^b^ (U/L)	18.00 (14.00; 24.00)	18.00 (14.00; 25.00)
GGT ^b^ (U/L)	15.00 (11.00; 21.00)	15.00 (11.00; 22.00)
AST ^b^ (U/L)	12.00 (10.00; 14.00)	12.00 (10.00; 15.00)
ALP ^b^ (U/L)	49.00 (42.00; 59.00)	49.00 (42.00; 59.00)
Observation Time ^b^ (years)	16.87 (16.12; 17.22)	16.88 (16.12; 17.24)
Age at Death ^b^ (years)	70.61 (60.36; 77.12)	68.97 (10.60)
Status ^a^		
Alive	1235 (82.00)	746 (82.20)
Dead	271 (18.00)	161 (17.80)
Cause of Death ^a^		
DSD or HCI-Related Mortality	20 (7.40)	11 (6.80)
CVD-Related Mortality	65 (24.00)	35 (21.70)
Cancer Mortality	102 (37.60)	68 (42.20)
Other Cause of Death	84 (31.00)	47 (29.20)
Smoking habit ^a^		
Never	1206 (80.10)	724 (79.80)
Current	300 (19.90)	183 (20.20)
Marital Status ^a^		
Single	101 (6.70)	60 (6.60)
Married or Cohabiting	1292 (85.80)	768 (84.70)
Separated or Divorced	32 (2.10)	24 (2.60)
Widower	81 (5.40)	55 (6.10)
Education ^a^		
Primary School	472 (31.30)	300 (33.10)
Secondary School	473 (31.40)	295 (32.50)
High School	363 (24.10)	208 (22.90)
Graduate	143 (9.50)	78 (8.60)
Illiterate	55 (3.70)	26 (2.90)
Dyslipidemia ^a^		
No	1132 (75.20)	683 (75.30)
Yes	374 (24.80)	224 (24.70)
Hypertension ^a^		
No	1031 (68.50)	612 (67.50)
Yes	475 (31.50)	295 (32.50)
Wine Consumption ^c^ (ml/die)	162.89 (210.51)	157.44 (209.28)
Beer Consumption ^c^ (ml/die)	55.07 (131.88)	51.61 (125.56)
Spirit Consumption ^c^ (ml/die)	62.49 (170.32)	54.18 (152.49)
Olive Oil Consumption ^c^ (gr/die)	26.90 (18.83; 38.00)	27.38 (18.83; 37.66)
Subtype 1 ^a^		
No	111 (7.40)	57 (6.30)
Yes	1395 (92.60)	850 (93.70)
Subtype 2 ^a^		
No	1435 (95.30)	872 (96.10)
Yes	71 (4.70)	35 (3.90)
Subtype 3 ^a^		
No	1346 (89.40)	872 (96.10)
Yes	160 (10.60)	35 (3.90)

SBP: Systolic Blood Pressure; DBP: Diastolic Blood Pressure; BMI: Body Mass Index; TG: Triglycerides; TC: Total Cholesterol; HDL-C: High-Density Lipoprotein Cholesterol; LDL-C: Low-Density Lipoprotein Cholesterol; ALT: Alanine Amino transferase; GGT: γ-Glutamyl transferase; AST: Aspartate Amino Transferase; ALP: Alkaline Phosphatase. DSD-Related Mortality: Digestive System Disease-related Mortality; HCI: Hepatic Cirrhotic Individuals; CVD-Related Mortality: Cardiovascular Disease-Related mortality; Subtype 1: Hepatic Steatosis and Overweight/Obesity; Subtype 2: Hepatic Steatosis Plus at Least Two Metabolic Abnormalities; Subtype 3: Hepatic Steatosis and Type 2 Diabetes Mellitus ^a^ Number (Percentage). ^b^ Median (IQR). ^c^ Mean ± (SD). Percentages Calculated for the Column.

**Table 2 jcm-13-01181-t002:** Cox Model of prognostic factors for the Survival of subjects with MAFLD (1000 bootstrap samples).

Prognostic Factors	Observed β	Bootstrap SE	Normal-Based 95% CI
Age at Enrollment (years)	0.146 **	0.012	0.122; 0.169
GGT (U/L)	0.009 **	0.002	0.004; 0.014
SBP (mmHg)	0.009 *	0.004	0.001; 0.017
Glucose (mg/dL)	0.004 *	0.001	0.001; 0.007
Gender (M vs. F)	0.356 *	0.135	0.091; 0.621
DBP (mmHg)	−0.016 *	0.008	−0.032; −0.001
TC (mg/dL)	−0.004 *	0.002	−0.008; −0.001
ALP (U/L)	0.008	0.004	−0.000; 0.016
Widowhood	0.550 **	0.162	0.232; 0.868

* *p*-value < 0.05; ** *p*-value < 0.001. GGT: γ-Glutamyl transferase; SBP: Systolic Blood Pressure; DBP: Diastolic Blood Pressure; TC: Total Cholesterol; ALP: Alkaline Phosphatase.

**Table 3 jcm-13-01181-t003:** Comparison of discriminant measures evaluated in the development and validation cohorts.

	R^2^_D_ ^a^	SE	Harrell’s C
Development			
Model ^b^	0.6845	0.03	0.8422
Validation			
Model ^b^	0.6930	0.03	0.8465

^a^ Explained Variation Statistics; ^b^ Cox model: Gender, Age at Enrollment, Widowhood, SBP, DBP, ALP, TC, Glucose and GGT, GGT: γ-Glutamyl transferase; SBP: Systolic Blood Pressure; DBP: Diastolic Blood Pressure; TC: Total Cholesterol; ALP: Alkaline Phosphatase.

## Data Availability

The datasets used and analysed in the current study can be obtained from the corresponding author, subject to reasonable request.

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
