# Peer review of "Development and Internal Validation of a Model for Predicting Overall Survival in Subjects with MAFLD: A Cohort Study"

_jcm, 2024, doi:10.3390/jcm13041181_

Round 1

Reviewer 1 Report

Comments and Suggestions for Authors

Bonfiglio et al. have presented a cohort study on the development and internal validation of a model for predicting overall survival in individuals with metabolism-dysfunction-associated fatty liver disease (MAFLD). Despite being a well-written critique, certain comments were made on the manuscript during the evaluation process. The following comments are to be addressed in the revised manuscript for further considering:

1.    As you have mentioned 15 hepatic cirrhotic individuals out of 20 DSD-related mortality, it would be more informative for the readers if could add this data under DSD in Table 1.

2.    The unit used for enzymatic values in Table 1 and other data presented by the authors are confused. In table 1, it was used as kat/L whereas in text it was written as U/L. It would be great if authors used SI units as international units for mentioning enzymes, e.g., IU/L.

3.    Please check the data set used in Table 1 to see if the comma, dot used for decimals are correct.

4.    Please add expansion for SBP, DBP, and more when it is being used in-text at the first instance.

5.    On page. 3, line 148, the authors have mentioned “……for each participant.” It should be corrected as “……from each participant….”.

6.    On page. 4, line 165, the word “current” should be replaced by current.

7.    Please check the continuous numbering for sub-sections in Materials and Method to avoid confusion.

8.    In page. 10, Line 331, the authors can use p<0.001 instead of pvalue<0.001.

9.    It would be appreciated if the authors could include some more concluding points from the results in the conclusions instead of saying a simple conclusion.

10. There are some general English grammar and typographical errors that should be checked and corrected throughout the manuscript.

Comments on the Quality of English Language

1.    There are some general English grammar and typographical errors that should be checked and corrected throughout the manuscript.

Author Response

Attached please find the revisions you requested

Reviewer 2 Report

Comments and Suggestions for Authors

The subject addressed in the manuscript is an interesting one, but a series of changes are necessary:

- the introduction is incomplete, it is necessary to present the risk actors associated with MAFLD and a prediction of their severity;

- methodology: it is necessary to present the principles of the protocol used in the development of the prediction model, objectives, predicted results

- the results and discussions must also be reported to the forecasts from the work protocol that was the basis for the development of the predictive model, but also the comparison with other predictive models used in the field

- the conclusions are much too narrow and do not contain a presentation of the added value brought by the study compared to other models used in such predictions

​

Author Response

(The authors gave the same response as above.)

Reviewer 3 Report

Comments and Suggestions for Authors

The authors developed a model to predict the survival probability of MAFLD patients and identified several prognostic factors. The manuscript is overall well presented with clear explanation of the methods and results. The authors have also honestly pointed out the potential limitation of their study. I have the following questions that I hope the authors could further clarify: 

1. The two cohorts were recruited at 1980s and 2000s, respectively. The 20 years difference could allow significant improvement in medical care conditions. Can the authors provide evidence that there is no big difference in the care protocol and thus potential improvement in medical care would not be a confounding factor, or discuss how this could impact the conclusion?

2. In Line 168, smoking habit of never or former has been combined into one group. I am slightly curious about this categorization as smoking history has been shown to contribute to various liver disease, including NAFLD (Rutledge et al, Smoking and Liver Disease, 2020). Can the authors divide the three groups in the model or provide justification? In the same paragraph, the word "current" and "day" were misspelled.

3. Has the authors ruled out the potential impact of HBV/HCV infection in the cohorts?

4. Are there censored data in Figure 2? Please mark those if applicable, or clarify in the text that there is no censored data points.

5. Overall mortality may be confounded by deaths irrelevant to liver disease, as pointed out by the authors that many died from cardiovascular disease or other causes. Selection of end point specifically related to MAFLD is more accurate. Competing risk analysis is also helpful to addressing this issue.

6. Based on the beta values obtained in the LASSO model, it looks like high total cholesterol reduces the death probability of MAFLD, which is against the assumption that high cholesterol is usually bad for health. Can the authors provide more discussion?

7. Figure 3: why can the observed event probability be larger than 1?

8. Figure 4 showed that the model is of predictive value in development cohort. How was the PI cutoff (8 and 10) determined? How many subjects are there in each groups? And how does the model perform in validation cohort? In line 328 the PI range is also labeled wrong, as it should be 8<PI<10 but not PI<8 and >10.

9. Is hemoglobin A1C level of the cohorts available to be included in the model? This is a more representative measurement of blood sugar level compared to blood glucose, as it reflects the average blood sugar level over the past 3 months.

10. The authors validated their model using a subset of the development cohort, but not a different cohort, which is a major concern to me. Can the authors refine their study by separating the development and validation cohorts, so that they include non-overlapping, different individuals?

Author Response

(The authors gave the same response as above.)

Round 2

Reviewer 2 Report

Comments and Suggestions for Authors

none

Author Response

Thank you very much for your comments. His request for " Can be improved" referring to the introduction and conclusion was made in the text and highlighted in green.

Reviewer 3 Report

Comments and Suggestions for Authors

The authors' response have addressed all my concerns. The predictive model developed in this manuscript will be very interesting for readers with a background in clinical medical research, especially in the MAFLD field. However the figures were not properly displayed in the revised manuscript.

Author Response

Thank you very much for your comments

I have uploaded high-definition images for the four figures in the manuscript into the text. I think, rightly, your criticism was referring to the low definition I had included in the text.